# Consumers’ Willingness to Pay for Rice from Remediated Soil: Potential from the Public in Sustainable Soil Pollution Treatment

**DOI:** 10.3390/ijerph19158946

**Published:** 2022-07-22

**Authors:** H. Holly Wang, Jing Yang, Na Hao

**Affiliations:** 1School of Public Affairs and China Academy of Rural Development (CARD), Zhejiang University, 866 Yuhangtang Road, Hangzhou 310058, China; wanghong@purdue.edu (H.H.W.); jingyoungnancy@gmail.com (J.Y.); 2Department of Agricultural Economics, Purdue University, 403 West State Street, West Lafayette, IN 47907, USA; 3Department of Economics, Beijing Technology and Business University, Fucheng Road #33, Beijing 100048, China

**Keywords:** heavy metal pollution, arable soil remediation, public participation, consumers’ willingness to pay, choice experiment, information intervention

## Abstract

Remediation of polluted soil on arable land is mostly funded by governments, with the understanding that the public’s willingness to pay for food produced on remediated soil can help establish a soil remediation model with more stakeholders. In contrast to previous studies that have focused on soil-remediation technologies’ diffusion, this study employs choice experiments to evaluate market preferences for crops grown from lands of varying quality that are reflected in consumers’ willingness to pay (WTP). The results show that consumers are willing to pay a small premium for rice labeled with remediated-soil claims, but the WTP for remediated-soil claim is less than that of an uncontaminated-soil claim. Consumers’ WTP for remediated-soil claim increases by 29.03% when combining with a well-known brand, and it increases by 71.17% when information is provided about the efficacy of cadmium and heavy-metal-pollution remediation; however, combining with the region-of-origin label does not increase WTP. We also find that, in early stages of promotion, online stores may reach target consumers more easily. Based on these results, we propose four implications for policymakers.

## 1. Introduction

Soil pollution has emerged as a public health threat in recent years. The contaminated condition of arable soil is of particular concern, as soil status is correlated with agricultural product safety [1,2]. Of all the arable land in China, 16% was found to suffer from heavy metal pollution at a level higher than the safe threshold, and 7% has excessive cadmium [3]. These toxic metals can enter food through agricultural production [4,5], causing human health problems [6,7,8]. Unfortunately, rice is a crop quite absorbent of cadmium and is the major crop grown in cadmium-polluted regions in China, the world’s largest rice producer, with 297 million hectares. This provokes rising public anxiety over the risks of excessive intake of cadmium over a prolonged period [9,10]. The Chinese government has introduced a series of environmental laws and regulations to improve the quality of arable soils, such as the *Soil Pollution Prevention Action Plan* and the *Soil Pollution Prevention and Control Law*. Moreover, many remediation projects are underway that are mainly funded by Chinese governments. International organizations also help. For instance, in 2017, the World Bank announced a US $100 million loan to ameliorate the heavy metal pollution of the rice farmland of over 8000 hectares in China’s Hunan Province. Presently, arable land that needs to be remediated in China consists of about 26.2 million hectares, and the corresponding restoration funding demand from the year 2021 to 2030 reaches about US $95 billion [11].

However, the current arable soil restoration projects are facing challenges in sustainability. In contrast to the widely perceived air and water pollution, soil pollution has received much less attention from the public, as soil quality is difficult to discern, and thus the effects of soil remediation programs are hard for the public to observe. This is partly because long-term accumulation is needed for the heavy metals to pass through the ecological chain to humans, so the consequences are not immediately recognizable. Although there are anxieties concerning food safety problems overall among consumers, the general public has a limited understanding of the importance of soil for food quality, not to mention that there is almost no direct access to quality information of the soil from where a crop is grown. Furthermore, currently there is no market mechanism to promote more entities to enter the arable soil remediation market. Many studies focus on exploring business models for non-arable soil remediation [12]. Among many arable-soil-remediation studies, most are centered on technology development and diffusion (see References [13,14,15] for examples), few include downstream entities such as food enterprises and consumers as stakeholders [16,17,18]. The existing business models of arable soil remediation overlook the welfare value brought to the public through improving food safety, which leads to missed opportunities for expanding the sustainable development of remediated land from the demand side.

For the sustainability of environmental protection, it is important to raise public awareness of arable-soil-remediation projects and involve them in such endeavors [19,20]. On the one hand, the public’s understanding of soil remediation can enhance its awareness of the relationship between soil quality and food quality and strengthen the supervision of the soil-remediation process, thus strengthening the effects of remediation. On the other hand, consumers’ preferences for crops grown from remediated soils may help support soil remediation by sharing the costs through purchases, which will attract more market entities to enter the remediation business model. Consumers are known to prefer pro-environmental and healthy food products, such as those labeled as green, organic, eco-friendly, or sustainable [21,22,23,24,25]. At the same time, individuals often suffer from food neophobia due to their lack of knowledge [26,27] or stigmatize safe foods simply because they were related to an environmental problem or once contaminated [28,29]. No study has examined consumer preferences for crops grown in soils that might be contaminated by heavy metals with or without remediation, and the effects of food neophobia or stigmatization on their preferences [30,31,32]. Therefore, it remains to be assessed the degree to which consumers value their benefits from crops grown on remediated soils.

Choice experiments are a stated preference method that has gained popularity when eliciting preferences for new or non-existent products [33]. With choice experiments and information treatment trials and using rice fields with cadmium pollution in China as the case, this study aimed to examine (1) how much premium consumers are willing to pay for rice grown from remediated land; (2) whether a difference exists in consumers’ willingness to pay (WTP) for rice from remediated land with cadmium-reduction treatment and that from uncontaminated land; and (3) what actions can be performed to help improve the marketization of agricultural products from remediated soils. Our study is the first to investigate potential public engagement in restoration projects by studying consumers’ WTP for rice from land of varying quality. The following sections present specific hypotheses we tested, detailed experiment designs, and the results of the tests, along with conclusions and implications for policymakers.

## 2. Hypotheses and Methodology

### 2.1. Hypotheses

The soil quality of the rice field can be seen as a credence attribute for rice that consumers and producers are asymmetrically informed of in the market. The quality of soils can be indicated with labels guaranteed by a private or public third party but regulated by the public sector or brands as distinctive signs and sources of trust [33]. Therefore, consumers would like to pay more for rice with a claim of safe heavy metal levels compared with rice without such a claim. In addition, although there is no difference in heavy metal content between remediated and uncontaminated soils, consumers may fear new technology or stigmatize of rice produced from remediated soils due to lack of cognition [34,35]. Based on the above discussion, we have the following two hypotheses:

**Hypothesis 1 (H1).** 
*Consumers would be willing to pay more for rice with a claim of safe heavy metal levels than for rice with no soil-quality claims.*


**Hypothesis 2 (H2).** 
*Consumers would be willing to pay more for rice grown in uncontaminated soils than for that grown in remediated soils.*


If the WTP difference exists, eliminating the negative stigma associated with remediated soils will be an important issue. In terms of enhancing consumers’ trust in the quality of crops grown from remediated soils, reducing consumers’ perceived risks, and mobilizing the general public, especially within the private sector, we propose four additional hypotheses.

Consumers’ understanding or perceptions of a particular technology can influence their assessment of food-safety risks and perceptions of food quality [36,37]. More information might help consumers reduce their anxiety about the risks of certain foods, leading to higher preferences for these foods [38,39]. There is evidence that the stigma attached to a food can be partially mitigated through behavioral interventions, such as exposure to information [40,41,42]. Therefore, we have the following hypothesis:

**Hypothesis 3 (H3).** 
*Consumers would be willing to pay more for rice grown in remediated soils if they have more information about the product.*


Brands can serve as another indicator of food quality [43,44]. Well-known brands may imply traceability [45] and more trustworthy food-safety standards [46,47]. Knowing this, we developed our fourth hypothesis:

**Hypothesis 4 (H4).** 
*Consumers would be willing to pay more for rice grown in remediated soils by well-known brands than that in remediated soils by ordinary brands.*


Regional reputation is an effective indicator of product quality in the lemon market [48]. Although China has a rice-production capacity of 209.61 million tons of rice each year [49], it still imports rice from countries such as Thailand, Vietnam, and the United States. However, the heavy metal contamination of soils and cadmium in rice have been frequently reported since 2013 for domestically produced rice, with nearly ten thousand articles in the leading media in China alone [50]. Reported heavy-metal-pollution cases have mostly appeared in Southern China, especially in Hunan and Hubei provinces, causing huge panic in consumers concerning heavy metal pollution in Southern China.

**Hypothesis 5 (H5).** 
*Consumers would be willing to pay more for rice grown in regions that have not suffered from severe heavy metal pollution than for rice grown in regions that have experienced this pollution.*


Rice produced in remediated soils involves new technologies that consumers do not know much about. Consumers using different purchasing channels may demonstrate different attitudes toward new technologies and therefore have different WTPs for the rice grown from remediated soils. China’s online retail market has developed very fast and has become the largest online retail marker in the world [51,52,53]. Wang et al. [54] found that innovative consumers preferred the use of online websites for both obtaining information and purchasing products, while conventional consumers preferred to use regular offline means for both obtaining information and purchasing products. We thus made the following hypothesis:

**Hypothesis 6 (H6).** 
*Consumers who are used to buying rice online would like to pay more for rice grown from remediated soils than those who are used to buying rice offline.*


### 2.2. Choice Experiment and Controlled Trials Design

The choice experiment simulates purchasing situations, and it forces respondents to really trade off one attribute against another [33]. To test the hypotheses above, an experiment was designed to obtain consumers’ WTPs for rice with different soil-quality claims and examine the influence of brands and regions on the WTPs. Meanwhile, a survey is distributed to all experiment participants. The experiment is used in tandem with the survey to explore the influence of consumers’ knowledge levels and consumption channels on their WTPs for rice with different soil-quality claims. All data were collected from an online platform.

Because the claims we are interested in have not been implemented in the rice market, the stated preference method is used to create a hypothetical market to understand preferences and behavioral logic. The choice experiment has been widely used for valuing actions on environmental protection or health-risk mitigation [55,56]. A hypothetical choice experiment is implemented to gauge consumers’ WTPs for certain rice attributes, including soil claims, brands, and regions of origin, which are directly used to test hypotheses H1, H2, H4, and H5. Consumers’ demographic indicators, purchasing channels, and knowledge levels of soil remediation obtained from the survey are used to test hypotheses H3 and H6.

Moreover, an information intervention is imposed in the choice experiment to better understand the impact of consumer knowledge on their WTPs (H3). Information intervention has been widely used to understand the impacts of consumers’ perceptions of certain attributes (for example, see References [57,58,59]). Participants are randomly divided into two groups: the control group in which participants directly take choice experiments, and the randomly selected treatment group in which information about cadmium rice is given before they take choice experiments. The provided intervention information includes the existence of soil heavy-metal pollution, health consequences of long-term consumption of cadmium-contaminated rice, and the remediation of soils polluted with heavy metals (see Appendix A for details).

A summary of the attributes included in the choice experiment is presented in Table 1. The price is included as the critical attribute needed to calculate the WTPs. Since the region of origin and varieties of rice in China are usually bound together, different levels of the region of origin also include the varieties of rice produced in the region in the experimental design. For price, we add a lower and a higher price choice together with the average price of 5 RMB/Jin. An opt-out option is included to better simulate a real-world-decision scenario [60].

The full-factorial design involves combinations of any two alternatives out of 3 × 3 × 5 × 2 = 90 possible distinct ones, totaling 4005 scenarios. A subset of 32 non-dominating scenarios is created by using a D-optimal design out of the full-factorial candidate set via a modified Federov search algorithm. Our design allows for the estimation of main effects and specific two-way interaction effects. To reduce the probability of respondent fatigue, the choice tasks are blocked into four groups, with each survey participant evaluating no more than eight scenarios. An example of one choice scenario is illustrated in Figure 1.

### 2.3. Econometric Methods

#### 2.3.1. RPL—Eliciting WTP for Different Attributes

Choice experiments are rooted in the Lancastrian consumer theory [61], which assumes that utility is derived from the characteristics of the goods or services. The random parameter logit model (RPL) is widely used in the analysis of discrete choice models in consumer economics [62] (Kilders and Caputo 2021). The RPL is chosen because it assumes consumers have heterogenous preferences. Following Train and Weeks [63], the utility level of the $j^{th}$ product in choice set *t* for the $n^{th}$ consumer can be expressed as follows:(1)$$\begin{array}{cc} U_{njt}=\sigma_{n}(optout & *nobuy+\left(-1\right)Price_{njt}+\alpha_{1n}Remediated_{njt}+\alpha_{2n}Uncontaminated_{njt}+\alpha_{3n}Brand_{njt}\\ & +\alpha_{4n}North_{njt}+\alpha_{5n}Huhu_{njt}+\alpha_{6n}Vietnam_{njt}+\alpha_{7n}US_{njt})+\varepsilon_{njt} \end{array}$$
where $\;\sigma_{n}$ is the price/scale factor equal to $\lambda_{n}/\upsilon_{n}$, with $\upsilon_{n}$ being the Gumbel distribution scale parameter for individual *n*, and $\lambda_{n}$ being the random coefficient of price (Scarpa et al., 2008); *nobuy* is equal to 1 for the no-buy option, and 0 otherwise; *Price* is a continuous variable represented by the experimental designed price levels; *Remediated* takes the value of 1 if there is a remediated soil claim, and 0 if otherwise; *Uncontaminated* takes the value of 1 if there is an contaminated soil claim, and 0 if otherwise; *optout* is the non-random coefficient representing the selection of no-buy option; $\alpha_{1n}$ to $\alpha_{7n}$ are the random coefficients representing the mean WTPs for certain attributes and are assumed normally distributed; and $\varepsilon_{njt}$ is the random error term following a type-one extreme value distribution.

The results of choice experiments provided the basis on which we tested all the hypotheses. The average WTP values for soil-quality-claim attributes help test H1, and the average WTP values for uncontaminated-soil claims are expected to be significantly positive. By comparing the average WTP difference between uncontaminated and remediated-soil claims, H2 was tested, and it is assumed that the average WTP for remediated-soil claims is significantly lower than the average for uncontaminated-soil claims. H4 and H5 were tested by comparing WTP values for the famous brand and region-of-origin attribute, respectively, and their interaction effects with soil-quality-claim attributes at various levels. Both the mean WTP for famous brands and the interacted WTP for famous brands and remediated-soil claims are expected to be significantly positive. The mean WTP for northeast-originated rice is also expected to be significantly positive, while the mean WTP for Hunan/ Hubei-originated rice is expected to be negative, given their accumulated quality reputation. Comparing the average WTP values between different information processing groups helps test H3. The mean WTP for remediated-soil claims in the information treatment group is expected to be higher than that of the control group. The individual consumer’s WTPs for a certain attribute serve as dependent variables for regressions tests of H3 and H6.

The region of origins only includes Chinese domestic regions. There are two reasons for this: firstly, rice exported to China from the United States, Vietnam, or Thailand has not yet suffered soil-remediation problems; secondly, this study aimed to propose suggestions on China’s soil-remediation projects. We also examined the interaction effects between the attributes. The utility function in (2) is further specified as follows:(2)$$\begin{array}{cc} U_{\mathrm{njt}}=\sigma_{n}(\mathrm{optout} & *\mathrm{nobuy}+(-1){\mathrm{Price}}_{\mathrm{njt}}+\alpha_{1n}{\mathrm{Remediated}}_{\mathrm{njt}}+\alpha_{2n}{\mathrm{Uncontaminated}}_{\mathrm{njt}}+\alpha_{3n}{\mathrm{Brand}}_{\mathrm{njt}}\\ & +\alpha_{4n}{\mathrm{North}}_{\mathrm{njt}}+\alpha_{5n}{\mathrm{Huhu}}_{\mathrm{njt}}+\alpha_{6n}{\mathrm{Vietnam}}_{\mathrm{njt}}+\alpha_{7n}{\mathrm{US}}_{\mathrm{njt}}\\ & +\alpha_{8n}{\mathrm{Remediated}}_{\mathrm{njt}}\times {\mathrm{Brand}}_{\mathrm{njt}}+\alpha_{9n}{\mathrm{Contaminated}}_{\mathrm{njt}}\times {\mathrm{Brand}}_{\mathrm{njt}}\\ & +\alpha_{10n}{\mathrm{Remediated}}_{\mathrm{njt}}\times {\mathrm{North}}_{\mathrm{njt}}+\alpha_{11n}{\mathrm{Contaminated}}_{\mathrm{njt}}\times {\mathrm{North}}_{\mathrm{njt}}\\ & +\alpha_{12n}{\mathrm{Huhu}}_{\mathrm{njt}}\times {\mathrm{Brand}}_{\mathrm{njt}}+\alpha_{13n}{\mathrm{Contaminated}}_{\mathrm{njt}}\times {\mathrm{Huhu}}_{\mathrm{njt}})+\varepsilon \mathrm{njt} \end{array}$$

Econometric estimations are carried out in Nlogit 6.0, which is often used for choice experiment analysis.

#### 2.3.2. OLS—Understanding Information Impacts and Consumer Heterogeneity

Hypotheses 3 and 6 were tested by considering the impact of information and consumption channels on consumers’ WTPs. Some independent variables to test the hypotheses include the rice-purchasing channels used by consumers, including online and offline channels. The other independent variables are the dummy variables of consumers’ knowledge level of soil remediation and the information intervention of soil remediation. Consumers’ reactions to the information intervention depend on their prior acceptance of the object [64]. To further understand the influence of information involvement, the interaction variables of information treatment with purchasing channels and information treatment with knowledge levels are added, respectively. Dependent variables are individual WTPs that are derived from RPL models without interaction terms in both control and information treatment groups. OLS regressions are used in Model (5):(3)$$indiv\_wtp_{ni}=\alpha_{0i}+\alpha_{1i}*CO_{n}+\alpha_{2i}*CO_{n}*inf+\alpha_{3i}*CH_{n}+\alpha_{4i}*CH_{n}*treat+\alpha_{5i}*FE_{n}+\epsilon_{ni}$$
where $indiv\_wtp_{ni}$ denotes the individual WTP for attribute *i* of consumer *n.* In our study, we mainly focus on WTP for rice with remediated-soil claims, and uncontaminated soil claim is used as a contrast. In Equation (3), the variable  treat is the dummy variable, which equals one if it belongs to the information treatment group, and it equals zero if it belongs to the control group; $CO_{n}$ denotes the knowledge levels of consumer *n*; $CH_{n}$ denotes the rice-purchasing channels of consumer *n*, including dummy variables representing fresh markets, specialty stores, small shops, supermarkets, and online markets; inf interacts with $CO_{n}$, as well as $CH_{n}$, to see the independent effect of information treatment on each variable. We also controled for fixed effects, $FE_{n}$, in some of the models such as city variables (dummy variables of all 20 cities) and individual demographic variables (variables in Table 2) to get robust results.

## 3. Data

### 3.1. Sample Overview

A national survey was administered to people in twenty cities in China in 2018. These cities represent first-tier municipalities, second-tier provincial capitals, and third-tier cities in both Northern and Southern China, including Beijing, Shanghai, Guangzhou, Shenzhen, Chengdu, Chongqing, Hangzhou, Suzhou, Tianjin, Wuhan, Nanjing, Qingdao, Changsha, Wuxi, Foshan, Ningbo, Zhengzhou, Shenyang, Yantai, and Dalian.

Figure 2 presents the distribution of samples in cities and the production of rice by province in China. The survey participants are screened to be adults (above 18 years of age), those who have purchased rice before, and those who have heard about cadmium. We employed the survey company Dynata to populate our survey to its national panel and collected 800 valid sample individuals, of which 396 participants are in the control group and 404 in the treatment group.

### 3.2. Data Description

Table 2 lists the summary statistics of demographic variables. The respondents have an average age of 34 years, and 49.50% are female. These statistics are representative of urban consumers in China who order groceries online (China Profile 2017). We also collected respondents’ education levels; that 80% of respondents have at least an associate’s degree. The large percentage of educated shoppers in this study indicates that most of them have sufficient knowledge and decision skills to evaluate the products in the choice experiments as part of the survey [65]. About two-thirds of the participants have a family annual income between 100 thousand and 500 thousand RMB. Balance tests were carried out following Blimpo (2014) [66] to see whether the characteristics of the control and treatment groups are the same. The *p*-values of balance tests are all insignificant in the last column in Table 2, which allows next comparisons between these groups.

In our survey, all participants have heard of cadmium rice, but their understanding of cadmium’s effects on health, its detection methods, and its importance to rice quality is different. Table 3 presents the percentage of different levels of knowledge by information treatment. About thirty percent of the participants report that cadmium pollution is not harmful to health; about ten percent of consumers believe that cadmium should be detected by using professional methods; and more than half say that they are concerned about the importance of soil quality to rice. China now has relatively advanced online shopping platforms, and consumers have formed the habit of searching for product information when buying goods online. With the development of urbanization, capital-controlled supermarkets and self-operated stores are also flourishing. Supermarkets are the most popular channel to buy rice, and the online channel has the lowest proportion. As for purchasing channels, the most popular channel to buy rice is the supermarket, which accounts for two-thirds of the total purchase, followed by specialty stores (nearly fifteen percent); other channels are wet markets (around 10%), small shops (about 10%), and online shopping (nearly 7%).

Participants are asked to rate the quality of rice produced in different regions on a scale of one to five before the choice experiments. The black bars in Figure 3 represent the mean scores. Rice from the northeast is perceived by consumers to be of higher quality, whether domestic or foreign. Rice from Southern China, including Hunan/Hubei provinces, is not highly rated by the respondents. This statistical result provides evidence for our fifth hypothesis (H5), confirming that Chinese consumers may judge rice quality by the production origin. They believe that rice from Southern China is not as good as rice from other regions.

## 4. Results

Econometric estimations were carried out in Nlogit 6.0. The coefficients were estimated directly in WTP space directly, as in Equations (1) and (2).

### 4.1. Positive WTP for Rice with Remediated-Soil Claims

In the control group (We also estimated joint models containing data from both the control and treatment groups by following Caputo (2020) [67] and the results are in Appendix B
Table A1. We rejected the null hypothesis that the preference structures are equivalent in the two groups with or without interaction terms at the *p* < 0.01, using the LR test (Chi-square = 61.31 without interaction terms and Chi-square = 126.19 with interactions terms). The results are shown in Appendix B
Table A1), choice experiment data of 396 individuals were used for the RPL estimation. The results are shown in Model 1 and Model 2 (see Table 4). The opt-out has a significantly negative WTP because rice is a staple food for Chinese consumers and making it unavailable will bring welfare loss to consumers.

The most noticeable WTP values in the control group are for the soil-quality attributes. Model 1 shows that consumers have a significantly positive mean WTP for rice with uncontaminated-soil claims (17.42 RMB/Jin). Model 2 shows that consumers have a positive mean WTP for both the single remediated-soil claim (5.02 RMB/Jin) and the single uncontaminated-soil claim (13.94 RMB/Jin). H1 was confirmed, as consumers have a great preference for rice produced from soil with documented quality of safe cadmium levels. The results also show that, without information disclosure, consumers have positive WTP for remediated-soil claims, but it is lower than that for rice with uncontaminated-soil claims. It can be inferred that, although soil quality meets the same safe cadmium level, consumers view rice with remediated- and uncontaminated-soil claims differently (Wald test results in Table 5 reject the null hypothesis that WTP for remediated claim equals WTP for the uncontaminated claim at the 1% level in both Model 1 and Model 2), thus confirming H2. This may be because consumers believe that there is a stigma attached to remediation technologies, or they underestimate the safety of the food produced in remediated soils.

The parameters for the standard deviations of WTPs are all significant, thus confirming our random parameter assumption to accommodate consumer preference heterogeneity, with only the exception of *Brand* in the model with interactions (Model 2). The magnitudes of the standard deviation are different across variables. Take Model 1 for example: the standard deviation for remediated is 5.03 RMB, and it is 4.25 RMB for uncontaminated, suggesting that consumers’ preference for remediated claim has more heterogenous distributions relative to the common safety claim. By comparing the standard deviations, we can also find that there are relatively smaller differences in consumers’ acceptance of well-known brands and rice from Hunan/Hubei and Vietnam.

### 4.2. Opposite Joint Evaluation Effects of Brand and Region of Origin with Remediated-Soil Claim

We now examine consumers’ mean WTP for a single *Brand* or *Region of Origin* attribute in the control group. The mean WTP for well-known brands is 5.72 RMB/Jin, indicating that Chinese consumers prefer rice with well-known brands. In the model, we used long-grain rice from Thailand as the base. As seen from the RPL model without interaction terms, the short-grain rice from Northeast China receives a positive WTP in price premium, long-grain rice from Vietnam and from China’s Hunan/Hubei region has an insignificant WTP, and short-grain rice from the US has a negative WTP. This is quite similar to the result of direct market observations and other food-preference studies that, in general, Chinese short-grain rice from its northeast regions is considered high-quality rice, and that Chinese consumers do not favor food commodities imported from the US over domestically produced ones [68,69].

Consumers differ not only in the mean WTP for remediated and uncontaminated-soil claims, but also in their joint evaluations when these two claims appear together with other attributes. The interaction terms in Model 2 present joint evaluation effects of brand, region of origin, and remediated-soil claims on WTP.

Well-known brands help significantly increase the WTPs for rice with both remediated and uncontaminated-soil claims, but the degree of increase is different. The coefficient of the interaction between well-known brands and remediated-soil claims is 1.44 RMB/Jin, and that between well-known brands and uncontaminated-soil claims is 0.58 RMB/Jin; both are significant at the 1% level. In terms of absolute size and relative percentage, the complementary effect between well-known brands and remediated-soil claims is much greater than that between brands and uncontaminated-soil claims (Wald test in Table 5 rejects the null hypothesis that the two complementary effects are equal at the 1% level).

The region of origin attribute decreases the WTP for remediated-soil claims but helps increase WTP for uncontaminated-soil claims. The coefficients of the interaction term between the region of origins and remediated-soil claims indicate that the origins and remediated-soil claims have substitution effects, no matter whether the participants prefer the rice from Northeast China (2.00 RMB/Jin decrease) or dislike the rice from Hunan/Hubei province (1.17 RMB/Jin decrease). The substitution effect of Northeast China is 0.86 RMB/Jin, higher than that of Hunan/Hubei province, and the Wald test (see Table 5) shows that the substitution effect of the two regions is significantly different at the 1% level. There is a complementary effect (0.62 RMB/Jin increase with northeast or 0.67 RMB/Jin increase with Hunan/Hubei province) between rice produce origins and uncontaminated-soil claims. However, the complementary effects are homogeneous across regions (see Wald test results in Table 5). Therefore, H4 is supported and H5 is partially supported since there is a substitution effect between remediated-soil claims and produce origins with a good or bad reputation.

The second and fourth columns in Table 6 present the conditional means of WTP for remediated- or uncontaminated-soil claims, given the influence of other attributes. Table 6 shows that there are greater variations between conditional WTPs for remediated-soil claims. For the rice with remediated-soil claims in the control group, well-known branded rice from the northeast region has the highest WTP (10.45 RMB/Jin), followed by rice with well-known brands (9.14 RMB/Jin); rice from Hunan/Hubei province without well-known brands has the lowest WTP (0.19 RMB/Jin).

According to the above findings, we can infer that well-known, more recognizable brand are more trustworthy to consumers if they claim to use soils with safe cadmium levels, and that brands can earn more trust if they use remediated soils and produce safer rice since consumers will see it as a sign of the enterprises’ social responsibility. Given the importance of project sustainability, we should be concerned about what to expect from firms; that is, a new orientation is needed if firms are expected to contribute to soil-remediation projects [70]. These brands should recognize the significant effect of environmental and social commitment on customer satisfaction [71]. For uncontaminated-soil claims, the brand or region of origin attribute improves consumers’ WTPs for uncontaminated-soil claims, with their influence being relatively monotonous. The opposite effect between the region of origin and the two soil-quality claims reminds us that consumers may have formed a habit of judging soil quality through product origins to select the quality of the rice as a result of information asymmetry. Thus, to some extent, the simple remediated-soil claim reminds consumers that the soil in that area is not good, as it used to be contaminated.

### 4.3. Positive Effect of Information Disclosure on WTP for Remediation-Soil Claims

The following part focuses on econometric results of the treatment group and the comparison between the treatment and control groups.

In the treatment group, choice experiment data collected from the 404 individuals who receive the information intervention are used in the same RPL estimations as in the control group. The results are shown in Model 3 and Model 4 (see Table 4). The distributions of individual WTP results of remediated and uncontaminated-soil claims are displayed in Figure 4.

The magnitude of standard deviations for all variables changed after information provision. The standard deviation for the remediated claim is 5.03 RMB in the control group, while it decreases to 3.51 RMB in the treatment group. Oppositely, the standard deviation for uncontaminated claim is 4.25 RMB in the control group, while it increases to 5.54 RMB in the treatment group. Similarly, the standard deviations for almost all other attributes increased. As shown in Figure 4, only the standard deviations of remediated-soil claims decrease in both RPL models, with or without interaction terms. The solid red bell-shaped curve is narrower and taller than the solid black curve, confirming that the information about cadmium rice reduces the heterogeneity of consumer preferences. The increase of standard deviations under information provision is consistent with findings of the majority of studies on information provision impacts on consumers’ WTP for certain attributes (see References [62,72], for example), since consumers react differently with the information, even if the information provided is not very relevant to the attributes [73]. As for how to explain the narrower distribution of WTP for remediated claim, which we care most about, we employ the theory of Fox et al. [74]. Specifically, almost all consumers do not have strong priors about the possible implications of the product with remediated claim; additional information about the credence attribute raises uncertainty and reduces the dispersion of WTPs.

Compared with the control group, the WTPs for remediated-soil claims and uncontaminated-soil claims increase in the treated group. As shown in Model 2 and Model 4, WTP for remediated-soil claims significantly increases by 71.17% (from 5.02 to 8.49 RMB/Jin); and the WTP for uncontaminated-soil claims increases slightly by 7.33% (from 13.94 to 15.09 RMB/Jin).

In the treatment group, the mean WTP for soil-claim attributes changes, and the WTP level of interaction terms of brand, region of origin, and soil claims also changes. As seen in Table 5, well-known brands help increase WTP for both remediated (0.76 RMB/Jin) and uncontaminated-soil claims (0.54 RMB/Jin). However, in contrast to the situation in the control group, the increases are the same at the level of 1%. In addition, it differs from the control group in that the substitution effects of region of origin with remediated-soil claims are homogeneous across regions (−1.30 RMB/Jin with northeast and −1.12 RMB/Jin with Hunan/Hubei). We assume that more information about remediated soils can alleviate consumers’ prejudice to some extent, regardless of the existing perceptions of remediated soil claim or stereotypes of regional soil quality.

The third and fifth columns in Table 6 present conditional mean WTP for remediated or uncontaminated-soil claims, given the influence of other attributes from the treatment group. Compared with the control group, consumers’ conditional WTP values for soil-quality claims increase, especially conditional WTP values for remediated-soil claims. The most obvious increase is WTP values for rice from Hunan/Hubei province with remediated-soil claims (7.24 RMB/Jin without famous brands compared with 0.19 RMB/Jin in the control group, and 10.37 RMB/Jin with famous brands compared with 4.57 RMB/Jin in the control group). After receiving information, the difference between the WTP for rice with different combinations of attributes becomes smaller.

Individuals’ WTP values for soil-quality attributes are obtained in Nlogit. OLS regressions are used to see whether there exists a significant impact from the information treatment, as well as from the influence of three aspects of knowledge levels (mentioned in Table 3). Results are shown in Models 5 and 6 and Models 9 and 10 in Table 7.

The coefficient of the information treatment variable *treat* is positive across all model specifications for the remediated-soil claims; it is significant in Models 5 and 6 and significant in Models 7 and 8, with all interaction terms with market channels when they enter the models. These statistics further confirm the results presented above. However, this variable is mostly insignificant for uncontaminated soil labels. The information treatment has a clear effect in telling consumers that remediated soils are safe and relieves their worries from not understanding soil remediation.

The coefficients of *proftest* are significantly positive, and its interaction term with *treat* is negative but not significant in Model 5. This shows that those consumers who believe that cadmium should be detected with professional methods have, on average, a 1.72 RMB/Jin higher WTP for remediated-soil claims than those who do not share the belief. However, we need to notice that the significance disappears when FE is added, meaning that this kind of phenomenon only occurs in certain groups.

The coefficients of *hharm* are insignificant in all regressions in Table 7, and its interactions with *treat* are significantly positive for uncontaminated-soil claims, while it ia negative for remediated-soil claims. We can infer that, when there is no additional information, consumers’ perceptions of health damage caused by cadmium will not influence their WTP for remediated or uncontaminated-soil claims. However, after consumers receive the information treatment, if consumers are aware of health risks caused by cadmium, they would be willing to pay at least 2.22 RMB/Jin more than those who are not aware for uncontaminated-soil claims in Models 9 to 12, and 1.76 RMB/Jin less for remediated-soil claims in Model 8 than those who are not aware. It seems that the information treatment does bridge the gap between those with and without cadmium health concerns for remediated soils but exaggerates the difference between the two groups of people for rice with uncontaminated-soil claims.

The coefficients of *soilq* and interaction terms with *treat* are all significant. Consumers who think that soil quality is of vital importance have lower WTP for remediated-soil claims while having a higher WTP for uncontaminated-soil claims compared with consumers who do not pay much attention to soil quality. However, information helps increase WTP for remediated-soil claims.

The consumers who view soil quality as important have a high WTP for uncontaminated-soil claims and have aversions to remediated-soil claims. However, the respondents raise their WTP after receiving information of cadmium rice. This indicates that raised awareness improves the participants’ WTP for remediated-soil claims. For consumers who believe that cadmium rice will bring health risks, additional information makes them more likely to lean toward uncontaminated-soil claims. It shows that, in the early stage of establishing consumers’ awareness of heavy metal pollution to soils, health and food safety are consumers’ primary concerns for rice consumption. All of these results show that the scientific information about soil remediation helps relieve consumers’ concerns and makes them more willing to support rice grown from such soils.

### 4.4. Screen Consumers through Buying Channels

Figure 5 presents the WTP for soil-remediation claims of consumers from different purchasing channels in the control group and the information treatment group. It can be intuitively observed that the average WTP of consumers from online purchasing channels is significantly different from that of consumers from other (offline) purchasing channels in both groups. Therefore, distinguishing rice consumers based on their online and offline consumption habits may be a possible way to find the target consumer group of soil-remediation claims.

To better study the heterogeneity of consumers’ WTP in different channels and the information impact on people’s WTP in different consumption channels, we ran Models 7 and 8 and Models 11 and 12. Table 7 presents the impacts of participants’ buying channels on their WTP for remediated or uncontaminated-soil claims, using the online channel as the base. As for WTP for remediated-soil claims, consumers who are used to the online channel have higher WTPs. After the information treatment, most of their WTPs significantly increase. In particular, the WTP of those who buy from specialty stores, wet markets, and small shops are significantly higher than their counterparts without the information treatment. These are reasonable because these three channels often have a lower quality reputation than supermarkets and online stores and suffer more from consumer WTP for remediated-soil claims. The information treatment brings them back more consumers than the other two channels. As for WTP for uncontaminated-soil claims, consumers accustomed to buying rice in supermarkets have significantly higher WTP (2.37 RMB/Jin) for uncontaminated-soil claims. However, with the cadmium-rice information intervention, the respondents’ WTP for supermarkets and small stores drops a lot.

The possible reason for these results is that the channel-specific consumers themselves have certain characteristics. Since the addition of fixed effects of variables does not affect the regression results, these characteristics cannot be explained simply by gender, age, education, or geographical differences. Consumers who buy rice in supermarkets may have more trust in them because supermarkets have helped them screen for a better variety of products. Online shoppers may have an independent ability to obtain information or have a higher expectation of information presented on the shopping platform, thus making them have a higher WTP for signals with a certain understanding and verification threshold. Of all the five channels to buy rice, the online approach has advantages in targeting people and providing expectations of information; thus, it is suitable for the new products to enter the market. However, offline channels also have certain advantages. After providing more information, offline consumers are more likely to form buying habits. Offline channels are thus suitable for promotion. This also validates H6.

## 5. Conclusions

### 5.1. Results Discussion

Arable soil remediation is not just related to crop yields; it is also related to the quality and safety of agricultural products. In this paper, we explore consumers’ WTP for rice grown from remediated soils, factors affecting their WTP, and possible factors that can accelerate or hinder the marketization of such agricultural products.

Chinese consumers have primarily associated food safety with the quality of soil, and there is a market for rice produced on remediated soils. Our study finds that about half of the survey participants think that soil quality affects the quality of rice, and that participants have a strong preference for rice grown on clean soils without pollution. These findings indicate a big concern in consumers for soil pollution affecting rice production. As such, the findings may motivate suppliers to pay more attention to soil quality and to seek production methods conducive to maintaining or improving soil quality. Consumers are willing to pay premiums for rice grown on remediated soils, especially as more information about the efficacy of cadmium rice and heavy-metal-pollution remediation becomes available.

It should be noted that consumers’ WTP for rice grown on remediated soils is less than that for rice grown on uncontaminated soils. Consumers’ WTP for rice grown on remediated soils is 0.48 RMB/Jin, while their WTP for rice grown on uncontaminated soils is 17.42 RMB/Jin. This may result in insufficient incentives for the supply side of soil-remediation production due to high remediation costs. Namely, food enterprises, investors, and farmers may choose to invest in areas with low risk of soil pollution and reduce their interest in soil-remediation projects, and this may result in a vicious circle of poor soil quality in this area or even in the surrounding areas. The positive side is that current stakeholders will work hard to protect their land from being contaminated. We should also be aware of the substitution effect of product origins and soil-quality labels. For rice of northeast origin, consumers’ WTP for rice from remediated soils decreases by 91.28%, and their WTP for rice from Hunan/Hubei province decreases by 31.52%. On the one hand, consumers may stigmatize products from areas where soil remediation exists, particularly in areas where they perceive the soil to be of higher quality. On the other hand, potential negative returns may weaken investment or remediation information disclosure in soil remediation by regional producers or governments that already have regional branding.

However, some initiatives can help improve consumers’ WTP for remediation claims from the perspective of enhancing consumer awareness and trust. Brands can effectively increase consumers’ trust in restoration claims. As can be seen from the results, Chinese consumers’ WTPs increase by approximately 29.03% if the remediation claim is combined with a well-known brand. Therefore, in the early stage of soil treatment, the participation of reputable enterprises other than the government or NGOs (non-governmental organizations) is needed to increase consumer trust and to open up the market. Big enterprises can capture the cash flow and gain social reputation, thus playing a role in investigating and supporting the supply chain, which can also enhance the marketization of the projects and boost the sustainability of soil remediation projects. Disclosing relevant information to the public helps raise public awareness of soil pollution and enhance consumers’ awareness of the effects of soil remediation in a proactive way. A piece of objective information increases WTP by approximately 71.17%, as found in the present study. Understanding restoration information improves people’s WTP. This is especially true for consumers who pay more attention to the impact of soil quality on food safety. Moreover, the increased disclosure of information can decrease consumers’ uncertainty and thus decrease WTP dispersion.

For marketers, our research also suggests possible solutions regarding resistance to the introduction of rice from remediated soils. Our results show that, in the early stage of promotion, online stores may have better access to target consumers. As consumers learn more about soil remediation, consumers in offline stores are likely to show greater loyalty to such products.

### 5.2. Policy Implications

Taking into account all the findings, this research has several implications for policymakers in governments regarding soil remediation. Firstly, stimulating consumer preferences can lead to higher social participation in soil remediation. Lack of public participation and funding are the two greatest difficulties with China’s current soil-remediation projects. This study proves that, from the perspective of consumer preferences, a path can be found to alleviate these two main points and update the current soil-remediation business model. Governments still play major roles in societal development, but in a way that requires increasing cooperation with the private sector [75]. Secondly, remediated-soil labeling can be explored in the current China, for which trust-building is the top priority. The impact of soil contamination on food safety has long been reflected in consumers’ choices of rice. The current food-labeling system has not eliminated consumers’ concerns about food safety, nor has it fully addressed consumers’ participation in improving farming conditions, such as soil remediation. Thirdly, though new policies need to be carefully specified, it is equally important to educate the public about soil pollution and restoration. Attention paid by consumers to farmland restoration is limited, not to mention that they have little knowledge about soil restoration. Therefore, even if Chinese consumers have realized the importance of soil quality for food safety, the absence of recognized science or media reports on soil remediation still will not allow consumers to participate in these projects in a significant way. Our study validates the role of soil-remediation information in mitigating the stigma surrounding remediation technologies and/or produce origins. Governments can share information about the positive effects of soil treatment to reduce public anxiety surrounding heavy metal pollution or regional discrimination. Detailed data on contaminated urban soils appear to be too difficult for researchers and the public to access in China nowadays [76], and more information about the importance of soil quality or the progress of restoration projects can be released. Lastly, because consumers’ WTP for price premium for rice from remediated soils over rice from contaminated soils is rather small compared to the costs of the projects, governments’ direct investment in such pollution cleanup projects is still necessary. Soil pollution is a result of long-term industrialization, which makes it very difficult to trace to specific polluters. It is even more difficult to hold them accountable and make such entities pay, not only because they may not have the financial capacity, but also because some of the damages took place long ago, even before legal regulations were established in China. Today, it is for the public’s benefit to clean up the damages and ensure the safety of rice production. It is not feasible for private enterprises to conduct huge environmental projects based on the small market price premium they may receive.

### 5.3. Limitations and Future Research Directions

This study has limitations. We investigated only rice, which is the staple food. Whether there are WTP differences with vegetables or traditional Chinese medicine grown from remediated soils still needs to be studied.

## Figures and Tables

**Figure 1 ijerph-19-08946-f001:**
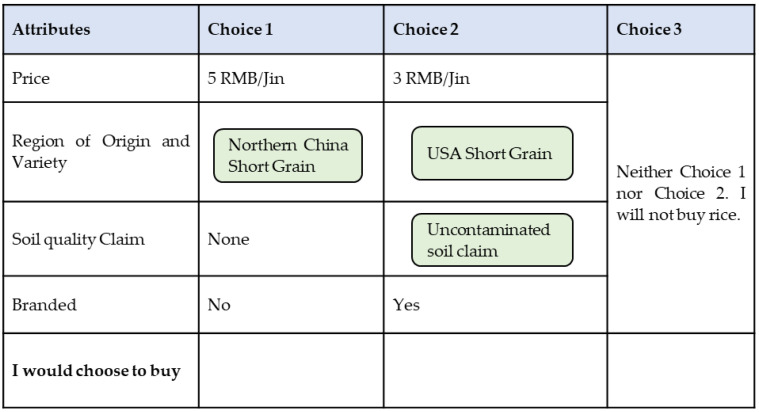
A sample of the choice sets presented to respondents.

**Figure 2 ijerph-19-08946-f002:**
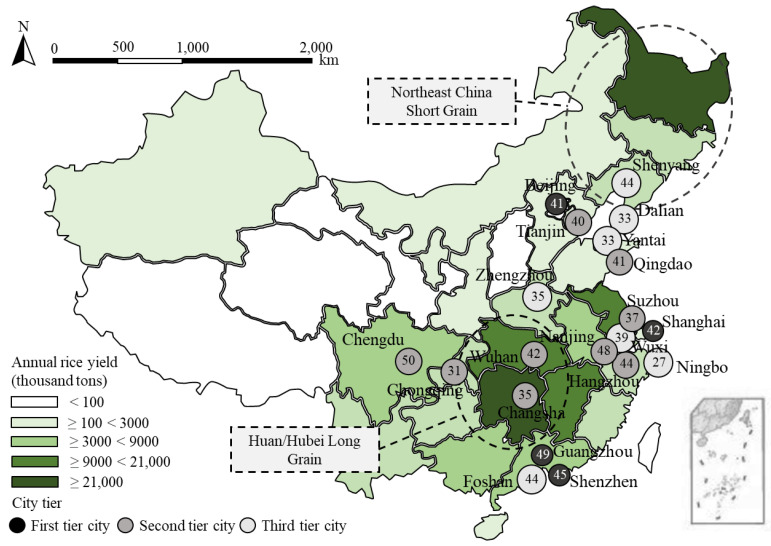
The production of rice by province in China and the distribution of samples in cities. Note: The number in the circle is the number of valid questionnaires collected from that city; the areas shown in dotted circles are the locations and names of the two rice-producing areas in China designed in our choice experiments.

**Figure 3 ijerph-19-08946-f003:**
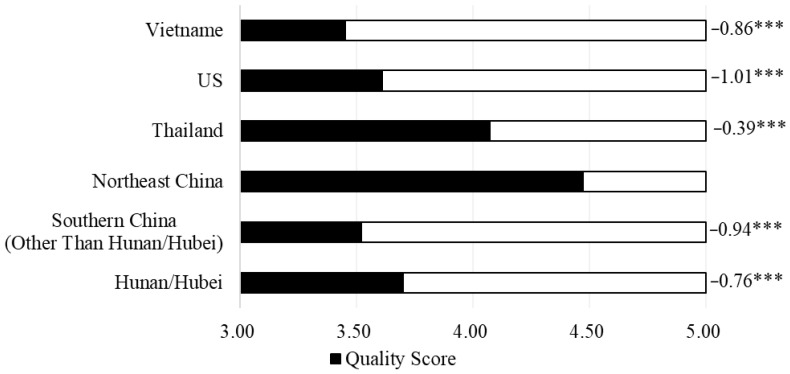
Quality scores by consumers for rice region of origin before the choice experiment. Note: The numbers and significance levels that follow the bar chart are the difference values between the region and the northeast rice score and the significant results of the Student’s *t*-test. *** Indicate *p* < 0.01.

**Figure 4 ijerph-19-08946-f004:**
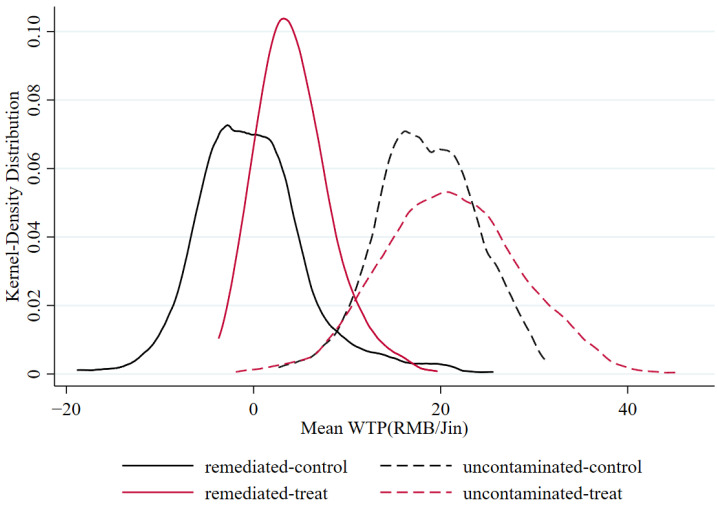
Distribution of WTP with and without information.

**Figure 5 ijerph-19-08946-f005:**
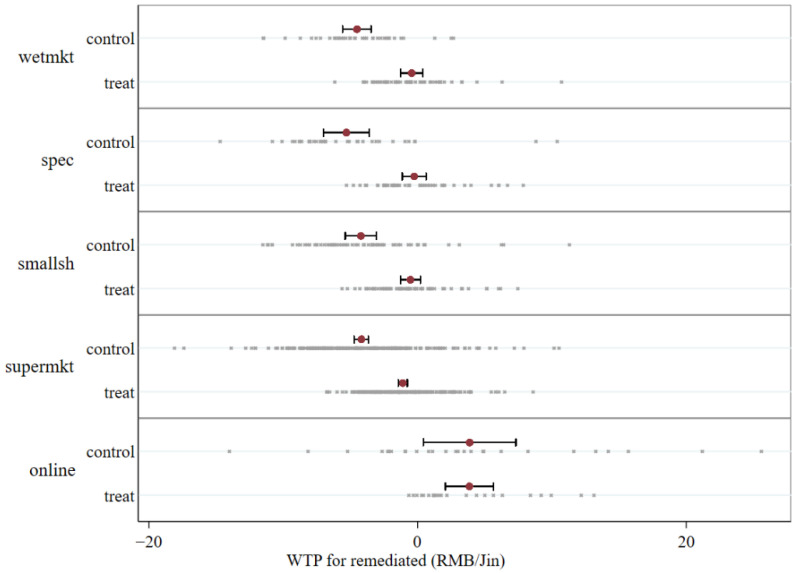
WTP for the remediated claim of different buying channels by treatment. Note: The red dot represents the mean WTP, the line represents 95% confidence interval, and a gray dot represents individual WTP.

**Table 1 ijerph-19-08946-t001:** Explanation of attributes designed in the choice experiment.

Attributes	Levels	Description
Price	3 RMB/Jin ^a^; 5 RMB/Jin; 10 RMB/Jin.	Average price.
Soil quality claim	Uncontaminated soil claim; Remediated soil claim; No claim ^b^.	Whether there is a claim for the soil qualified for safe cadmium level or remediated from previous contamination.
Region of origin and variety	Vietnam long grain; USA short grain; Thailand long grain ^b^; Northern China short grain; Huan/Hubei long grain.	Country and region of origin, each with only one dominating variety.
Brand	Famous brand; Not famous brand ^b^.	Whether it is a recognizable famous brand.

Note: ^a^ One Jin is five hundred grams. ^b^ Underlined is the base level.

**Table 2 ijerph-19-08946-t002:** Summary of sample characteristics.

	Description	Full Sample	Control	Treat	*p*-Value ^a^
Observation	Number of participants	800	396	404	
Age	Average age (year)	34.24	34.58	33.91	0.31
Gender	Male (%)	50.50	50.51	50.50	1.00
	Female (%)	49.50	49.49	49.50	
Educational Level	High-school education (%)	18.63	18.43	18.81	0.93
	Undergraduate education (%)	74.63	74.49	74.75	
	Graduate/professional (%)	6.75	6.82	6.19	
Family Annual Income	Under 50,000 RMB (%)	3.75	3.79	3.71	0.75
	50,001–100,000 RMB (%)	14.63	15.40	13.86	
	100,001–200,000 RMB (%)	33.63	43.69	45.54	
	200,001–500,000 RMB (%)	33.00	33.59	32.43	
	Over 500,000 RMB (%)	4.00	3.54	4.46	

Note: ^a^ The *p*-value refers to the balance test of the equality of the two groups.

**Table 3 ijerph-19-08946-t003:** Proportion of knowledge levels and buying channels.

Variables	Coding and Descriptions	Control (%)	Treat (%)	*p*-Value ^a^
Knowledge: Who believed that				
hharmq	=1 if cadmium pollution will cause health damage	75.51	75.50	0.21
	=0 if it will not lead to health damage	24.50	24.49	
proftest	=1 if cadmium rice can be detected using professional methods	9.34	6.93	1.00
	=0 if can be detected without using a professional method	90.66	93.07	
soilq	=1 if soil is of vital importance to rice quality	52.27	55.69	0.33
	=0 if is not important to rice quality	47.73	44.31	
Buying channels:				
wetmkt	=1 if usually buy rice in wet market; =0 otherwise	9.60	10.88	0.64
specs	=1 if usually buy rice in specialty stores; =0 otherwise	8.84	10.50	
smallsh	=1 if usually buy rice in small shops; =0 otherwise	14.90	14.88	
supermkt	=1 if usually buy rice in supermarkets; =0 otherwise	59.60	57.25	
online	=1 if usually buy rice in online stores; =0 otherwise	6.82	6.25	
Obs.		396	404	

Note: ^a^ The *p*-value refers to the balance test of the equality of the two groups.

**Table 4 ijerph-19-08946-t004:** RPL (WTP space) results of both control and treatment groups with or without interaction terms.

Variables/ Interactions	Control				Treat			
	Model 1		Model 2		Model 3		Model 4	
	Mean	SE	Mean	SE	Mean	SE	Mean	SE
**Mean estimation**								
**Random parameter**								
Price	1	0	1	0	1	0	1	0
Remediated	0.48	0.58	5.02 ***	0.71	4.22 ***	0.59	8.49 ***	0.72
Uncontaminated	17.42 ***	0.63	13.94 ***	0.58	20.70 ***	0.78	15.09 ***	0.67
Brand	5.72 ***	0.31	2.94 ***	0.30	5.35 ***	0.34	2.36 ***	0.34
North	3.86 ***	0.43	2.20 ***	0.47	4.19 ***	0.47	2.47 ***	0.46
Huhu	0.42	0.37	−3.69 ***	0.42	−0.59	0.44	−0.12	0.45
Vietnam	0.57	0.40	0.36	0.40	0.49	0.40	1.82 ***	0.38
US	−3.53 ***	0.53	−2.28 ***	0.48	−2.89 ***	0.51	−0.17	0.45
**Non-random parameter**								
Brand * Remediated			1.44 ***	0.14			0.76 ***	0.15
Brand * Uncontaminated			0.58 ***	0.08			0.54 ***	0.08
North * Remediated			−2.00 ***	0.14			−1.30 ***	0.13
North * Uncontaminated			0.62 ***	0.10			0.66 ***	0.10
Huhu * Remediated			−1.17 ***	0.17			−1.12 ***	0.17
Huhu * Uncontaminated			0.67 ***	0.10			0.10	0.11
Optout	−2.45 ***	0.13	−2.42 ***	0.24	−2.52 ***	0.14	−2.69 ***	0.25
**SD estimation**								
Price	0		0		0		0	
Remediated	5.03 ***	0.55	5.10 ***	0.56	3.51 ***	0.50	0.14	0.51
Uncontaminated	4.25 ***	0.51	4.00 ***	0.45	5.54 ***	0.56	5.26 ***	0.49
Brand	1.08 ***	0.37	0.28	0.38	3.18 ***	0.38	2.75 ***	0.34
North	4.68 ***	0.47	4.63 ***	0.44	4.93 ***	0.54	4.29 ***	0.44
Huhu	2.52 ***	0.48	2.57 ***	0.47	4.37 ***	0.49	3.51 ***	0.43
Vietnam	2.01 ***	0.39	2.99 ***	0.40	2.47 ***	0.45	0.20	0.38
US	6.22 ***	0.54	5.07 ***	0.54	4.60 ***	0.51	4.08 ***	0.50
**Model statistics**								
AIC	5042.70		4963.10		5215.90		5197.50	
Log likelihood	−2504.35		−2458.57		−2590.96		−2575.76	
Choices	396 × 8		396 × 8		404 × 8		404 × 8	

Note: SE: standard error; SD: standard deviation; *** and * indicate *p* < 0.01 and *p* < 0.1, respectively; AIC, Akaike information criterion.

**Table 5 ijerph-19-08946-t005:** Wald test results.

	Control		Treat	
	Model 1	Model 2	Model 3	Model 4
Remediated vs. Uncontaminated	−16.96 *** a	−8.92 ***	−16.48 ***	−6.60 ***
	[−19.68, −14.22] b	[−11.68, −6.14]	[−19.38, −13.56]	[−9.29, −3.90]
Brand * Remediated vs. Brand * Uncontaminated		0.86 ***		0.22
		[0.26, 1.46]		[−0.40, 0.84]
Remediated * North vs. Remediated * Huhu		−0.86 ***		−0.18
		[−1.45, −0.26]		[−0.70, 0.34]
Uncontaminated * North vs. Uncontaminated * Huhu		−0.04		0.56 ***
		[−0.46, 0.38]		[0.16, 0.96]

Note: ^a^ the mean WTP difference between the former and the latter attributes; *** and * indicate *p* < 0.01 and *p* < 0.1, respectively; ^b^ square brackets indicate 95% confidence intervals.

**Table 6 ijerph-19-08946-t006:** WTP for soil claims with consideration of brand and region-of-origin attributes.

RMB/Jin	Remediated		Uncontaminated	
	Control	Treat	Control	Treat
Only soil claim	5.02 *** a	8.49 ***	13.94 ***	15.09 ***
	[2.25, 7.81] b	[5.68, 11.32]	[11.66, 16.22]	[12.46, 17.70]
+Well-known brand	9.14 ***	11.62 ***	17.46 ***	18.00 ***
	[6.37,12.45]	[8.59, 14.64]	[14.65,20.27]	[14.93, 21.06]
+Northeast	5.21 ***	9.66 ***	16.76 ***	18.23 ***
	[2.18, 8.25]	[6.56, 12.77]	[13.51,20.02]	[14.59, 21.87]
+Hunan/Hubei	0.19	7.24 ***	10.92 ***	15.07 ***
	[−2.21, 2.59]	[4.49, 9.99]	[8.37, 13.46]	[11.81, 18.32]
+Northeast +Well-known brand	10.45 ***	12.96 ***	20.30 ***	20.56 ***
	[7.01, 13.90]	[9.44, 16.49]	[16.28, 24.16]	[16.41, 24.75]
+Hunan/Hubei +Well-known brand	4.57 ***	10.37 ***	14.43 ***	17.98 ***
	[1.89, 7.24]	[7.23, 13.50]	[11.38, 17.49]	[14.20, 21.75]

Note: ^a^ mean WTP; *** indicates *p* < 0.01; ^b^ 95% confidence intervals.

**Table 7 ijerph-19-08946-t007:** OLS interaction item of regression results 1.

Variables	WTP for Remediated-Soil Claims				WTP for Uncontaminated-Soil Claims			
	Model 5	Model 6	Model 7	Model 8	Model 9	Model 10	Model 11	Model 12
treat	4.50 ***	4.55 ***	0.31	0.49	0.01	0.16	5.41 ***	3.16
	(0.76)	(0.76)	(1.36)	(1.52)	(0.94)	(0.95)	(1.76)	(1.92)
proftest	1.72 **	1.07		0.66	0.04	0.41		0.54
	(0.85)	(0.86)		(0.86)	(1.05)	(1.09)		(1.09)
hharm	0.77	0.70		−0.73	−0.15	0.04		−0.92
	(0.59)	(0.59)		(1.28)	(0.73)	(0.74)		(1.62)
soilq	−1.82 ***	−1.73 ***		0.78	2.70 ***	2.66 ***		0.16
	(0.50)	(0.50)		(0.59)	(0.63)	(0.64)		(0.74)
treat * proftest	−1.28	−1.19		−0.95	−0.82	−0.66		2.09 **
	(1.28)	(1.28)		(0.82)	(1.59)	(1.62)		(1.04)
treat * hharm	−0.98	−1.00		−1.76 ***	2.31 **	2.22 **		2.50 ***
	(0.82)	(0.82)		(0.50)	(1.02)	(1.04)		(0.64)
treat * soilq	1.47 **	1.42 **		1.43 **	1.37	1.18		1.30
	(0.71)	(0.71)		(0.70)	(0.88)	(0.89)		(0.89)
wetmkt			−4.34 ***	−3.54 ***			−0.06	−0.83
			(1.22)	(1.25)			(1.58)	(1.63)
specs			−5.38 ***	−5.09 ***			0.92	0.81
			(1.24)	(1.25)			(1.60)	(1.64)
smallsh			−3.94 ***	−3.39 ***			1.33	0.91
			(1.12)	(1.14)			(1.45)	(1.49)
supermkt			−3.89 ***	−3.20 ***			2.37 *	2.14 *
			(0.98)	(0.99)			(1.26)	(1.30)
treat * wetmkt			5.13 ***	5.05 ***			−1.16	−1.42
			(1.72)	(1.71)			(2.23)	(2.24)
treat * specs			6.42 ***	6.81 ***			−2.08	−2.36
			(1.74)	(1.74)			(2.25)	(2.27)
treat * smallsh			4.61 ***	4.70 ***			−4.20 **	−4.48 **
			(1.63)	(1.63)			(2.11)	(2.13)
treat * supermkt			3.81 ***	3.80 ***			−3.21 *	−3.49 *
			(1.43)	(1.44)			(1.86)	(1.88)
Controls								
City FE	N a	Y	N	Y	N	Y	N	Y
Individual FE	N	Y	N	Y	N	Y	N	Y
Constant	0.06	0.34	3.65 ***	3.31 **	17.30 ***	17.75 ***	16.92 ***	16.80 ***
	(0.53)	(1.37)	(0.92)	(1.61)	(0.66)	(1.72)	(1.20)	(2.03)
Obs	800	800	800	800	800	800	800	800
R-squared	0.18	0.23	0.19	0.25	0.13	0.15	0.06	0.17

Note: ***, **, and * indicate *p* < 0.01, *p* < 0.05, and *p* < 0.1, respectively. Standard error in parentheses (). ^a^ N (no) indicates without certain FE (fix effect); Y (yes) indicates with certain FE.

## Data Availability

The data presented in this study are available upon request from the corresponding author.

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
