# Peer review of "Consumers’ Willingness to Pay for Rice from Remediated Soil: Potential from the Public in Sustainable Soil Pollution Treatment"

_ijerph, 2022, doi:10.3390/ijerph19158946_

Round 1

Reviewer 1 Report

Review on article entitled: “Consumers’ Willingness to Pay for Rice from Remediated Soil: Potential Partnership between Private Parties and the Government in Sustainable Soil Pollution Treatment”

This paper explores the willingness of Chinese consumers to pay for rice that comes from remediated soil and uses a choice experiment to determine the factors affecting this willingness. The paper is very well presented, especially the methodology part where the different hypotheses are described and justified based on the literature. Moreover, the different models are also explained based on the hypotheses. The presentation of the results is also linked to the six hypotheses, which makes the manuscript easy to follow.

There are a few things that I think need to be clarified before the paper is published.

The title does not really correspond to what is presented in the paper. I do not think it is necessary to include Potential Partnership between Private Parties and the Government in Sustainable Soil Pollution Treatment, because this leads the reader to think that this has been actually investigated.

The abstract is well presented, concise including all the relevant information to attract readers attention.

The introduction is concise but includes enough background information to set the scene.

Methods: In relation to the methods, it is general practice to describe the software used for the data collection and for the analysis of the data, which I could not find in the paper. Nlogit is mentioned in the results section but it should be mentioned in the methods as well.

Moreover, the choice experiments need more description on how they were actually conducted. The tables are very useful but description in the text is necessary to give details of how the tests were conducted in reality.

Results: The results are presented following the hypotheses sequence and overall it is easy to follow the train of through of the authors. Some minor consideration in the following tables. However a discussion part is important to be included and this cannot be found in the paper.

Table 3: Be more descriptive in legend. In the same table the autros refer to balance test, but this was not described in the methods. The same test is mentioned in table 2, but no reference is provided to the test.

Table 4, Table 5, and Table 7:  please include more information on the legends of these tables. They need to be more descriptive and the abbreviations need to be explained.

Lines 443-445: Please consider rewording the text here to make it easier for the reader.

Discussion: there is no discussion of the results and linking of the findings with the papers mentioned in the methods and the introduction.

Conclusions: are well presented with reference to the aims of the work but it is best to change the order in the lines 551-552. The factors that can accelerate or hinder marketization could be at the end of the sentence since they are discussed at the end of the conclusions.  

Line 607: change pain with main

Could the authors comment on the method used (choice experiments) and how this is linked or compared with other methods employed in the past to explore ?

Reviewer 2 Report

The second paragraph of the introduction (lines 46-62) clearly establishes the need for this research, explaining the public's lack of knowledge of soil pollution risks, and the existing lack of research on stakeholder perceptions of soil remediation. 

The introduction also establishes that increased public willingness to buy products grown in remediated soil is crucial for the sustainability of soil remediation programs, which is why public perceptions should be identified through research.

The authors indicated unambiguously that there is a  need to establish markets for agricultural products from remediated soil, and stakeholder perceptions are a crucial component of establishing such markets..

Section 2, Hypotheses and methodology begins with a formulation of six hypotheses about the consumer's preferences for rice produced in remediated and unpolluted soils. The hypotheses are clearly conceived and relate clearly to the behavioral intentions of the consumer, the stigma attached to products grown in polluted and/or remediated soil, and the importance of information about the product’s quality.

The validity of the research outcome is good, because the authors have formulated the key hypotheses correctly. The authors formulated the hypotheses to test the impact of claims about the product’s putative effect on human health. It is crucial that the research is not testing the perceptions of the soil remediation technology per se, but in fact focuses on testing the impact of claims (advertisers’ claims, producers’ claims) on the consumer’s perceptions and willingness to pay.

In section 2.2, Choice Experiment and Controlled Trials Design, the authors describe the research procedure. It is a relatively complicated procedure, involving not only a survey, but also a choice experiment. The choice experiment divides research subjects into two groups, one of which receives information about soil pollution and remediation prior to the experiment.

In designing the choice experiment, the authors incorporated local preferences for brands and region of origin, which is a significant area of consumer knowledge in choosing rice for puchase. This use of local knowledge for conceptualizing consumer behaviors adds to the validity of the measurement of willingness to pay.

The sample of research subjects was nationwide urban consumers of rice and seems to be properly screened for the relevant individuals. The sample is relevant in terms of gender and average age, and the authors say it represents the average person who would purchase foodstuffs through internet shopping.

 The results section gives a clear explanation, beginning with the WTP for remediated soil claims, but also continuing to explain the differences in WTP when claims for soil quality appear together with other attributes.

In particular, the authors discovered that region of origin for the rice decreases the WTP for remediated soil claims but increases it for uncontaminated soil claims. This distinction is very important for future efforts to market rice from remediated soils.

The detailed discussion in section 4.3 Positive Effect of Information Disclosure on WTP for Remediation Soil Claims gives much credibility to the authors' assertion that the hypothesis #3 was supported.

  The fact that the authors included a comparison of the consumer's willingness to pay in different consumption channels--online and offline--adds to the usefulness of this study for policy applications. FIrst, it draws the discussion to the cultural and social characteristics of the consumers themselves, specifically their willingness to seek out and evaluate information. This ties the discussion to a wider body of studies on risk perception, which have addressed the process referred to as the personalization of risk. Second, in a practical sense, this differentiation of consumers in online and offline purchasing channels suggests in a practical manner which channels need to be targeted for promoting the value of rice from rehabilitated soils.

The concluding section (lines 549-636) provides a succinct and direct statement of the major findings of this research. The fact that Chinese consumers associate the quality of the agricultural product with the quality of the soil was confirmed; furthemore, the study found that Chinese consumers are willing to pay a premium for rice grown in remediated soils.

The concluding section also provides a practical discussion of the ways that consumer trust can be consolidated for products grown in remediated soil.

The section concludes that the willingness to pay also provides a valid reason for the government's direct investment in soil remediation, and argues that it should be continued for its public benefit.

Author Response

Thanks! We appreciate the detailed comments.

Reviewer 3 Report

The paper “Consumers’ Willingness to Pay for Rice from Remediated Soil: Potential Partnership between Private Parties and the Government in Sustainable Soil Pollution Treatment” deals with very interesting and important topic. I have a few minor comments to correct in the text of the manuscript:

  • Figure 1 - to me this is a table not a drawing, please correct this.
  • The manuscript is missing a discussion. The authors should discuss the results and how they were interpreted in the perspective of previous studies. The results of the study should also be related to the studies of other authors.
  • Conclusions should include highlight any new findings, and explain how the work could be extended in the future. In this section you should also clearly present the following important aspects:

- the research limitations;

- the future research directions.

  • Some literature items in the bibliography should be corrected according to the publisher's guidelines (e.g. 11, 46).

Round 2

Reviewer 3 Report

The new version of the manuscript incorporates the reviewer's comments